# Novel *JAG1* Deletion Variant in Patient with Atypical Alagille Syndrome

**DOI:** 10.3390/ijms20246247

**Published:** 2019-12-11

**Authors:** Emanuele Micaglio, Andreea Alina Andronache, Paola Carrera, Michelle M. Monasky, Emanuela T. Locati, Barbara Pirola, Silvia Presi, Mario Carminati, Maurizio Ferrari, Alessandro Giamberti, Carlo Pappone

**Affiliations:** 1Arrhythmology and Clinical Electrophysiology Department, IRCCS Policlinico San Donato, 20097 San Donato Milanese, Milan, Italy; Emanuele.micaglio@grupposandonato.it (E.M.); Michelle.monasky@grupposandonato.it (M.M.M.); EmanuelaTeresina.Locati@grupposandonato.it (E.T.L.); 2Department of Congenital Heart Surgery, IRCCS Policlinico San Donato, 20097 San Donato Milanese, Milan, Italy; Andreea.andronache@gmail.com (A.A.A.); Mario.carminati@grupposandonato.it (M.C.); Alessandro.giamberti@grupposandonato.it (A.G.); 3Division of Genetics and Cell Biology, Unit of Genomics for diagnosis of human diseases IRCCS San Raffaele Sci. Inst., 20132 Milan, Italy; Carrera.paola@hsr.it (P.C.); magenfe@yahoo.com (M.F.); 4Laboratory of Clinical Molecular Biology and Cytogenetics, IRCCS San Raffaele Sci. Inst., 20132 Milan, Italy; Pirola.barbara@hsr.it (B.P.); Presi.silvia@hsr.it (S.P.); 5San Raffaele Vita-Salute University, 20132 Milan, Italy

**Keywords:** Alagille syndrome, JAG1, genetic testing, heart, cardiac, liver, jaundice, neural tube defect, variant, mutation, bilirubin, cholestasis, cyanosis, hypoplasia, pulmonary branches

## Abstract

Alagille syndrome (AGS) is an autosomal-dominant disorder characterized by various degrees of abnormalities in the liver, heart, eyes, vertebrae, kidneys, face, vasculature, skeleton, and pancreas. This case report describes a newborn child exhibiting a congenital neural tube defect and peculiar craniofacial appearance characterized by a prominent forehead, deep-set eyes, bulbous nasal tip, and subtle upper lip. Just a few hours after birth, congenital heart disease was suspected for cyanosis and confirmed by heart evaluation. In particular, echocardiography indicated pulmonary atresia with ventricular septal defect with severe hypoplasia of the pulmonary branches (1.5 mm), large patent ductus arteriosus and several major aortopulmonary collateral arteries. Due to the association of peculiar craniofacial appearance and congenital heart disease, a form of Alagille syndrome was suspected. In addition, on the fifth day after birth, the patient developed jaundice, had acholic stools, and high levels of conjugated bilirubin and gamma-glutamyltransferase (GGT) were detected in the blood. Genetic testing revealed the novel variant c.802del in a single copy of the *JAG1* gene. No variants in the *NOTCH2* gene were detected. To the best of our knowledge, this is the first clinical description of a congenital neural tube defect in a molecularly confirmed Alagille patient. This work demonstrates a novel pathogenic heterozygous *JAG1* mutation is associated with an atypical form of Alagille syndrome, suggesting an increased risk for neural tube defects compared to other Alagille patients.

## 1. Introduction

Alagille syndrome (AGS) is an autosomal-dominant disorder characterized by various degrees of abnormalities in the liver, heart, eyes, vertebrae, kidneys, face, vasculature, skeleton, and pancreas [1,2]. The estimated frequency is one in 30,000 live births [3]. Alagille syndrome has been defined as the paucity of the interlobular bile ducts together with three to five of the following: chronic cholestasis, cardiac disease, skeletal abnormalities, ocular abnormalities, and a characteristic facial phenotype [4].

The syndrome received its name from the French pediatrician Daniel Alagille (1925–2005), who first described it in 1969, focusing on hepatic pathologies [5]. Since this discovery, many other authors characterized AGS cases by hepatic involvement and various other congenital abnormalities, all with the same genetic causes [6].

The diagnosis of AGS is difficult because of the variability in the clinical presentation, even between family members, despite almost complete penetrance [2,4]. However, genetic testing can aid in the diagnosis, as the majority of cases are due to haploinsufficiency of the *JAG1* gene, mapping on the short arm of chromosome 20 (20p11.2–20p12), either because of point mutations, exon and whole gene deletions or the microdeletion of 20p12 [7]. *JAG1* mutations account for around 94% of cases, while *NOTCH2* mutations are less common [1]. Both of these genes are involved in the Notch signaling pathway and play an important role in cell fate determination [8]. New mutations commonly occur (about 60%), and the rate of germline mosaicism may also be relatively high [7]. Prenatal testing is not able to predict the severity of the phenotype because of a reduced penetrance (47% [9]). Due to the variability in clinical presentation, early genetic testing is needed to confirm the syndrome so that preventative measures can be taken to prevent complications in multiple organs.

## 2. Case Presentation

Written informed consent of human subjects (or parents, in the case of minor children) included in this case series report was obtained for their participation in the study and for publication of this case report. The procedures employed were reviewed and approved by the local Ethics Committee.

The proband is a female child who was born in Romania from healthy, non-consanguineous parents with no remarkable medical history at 37 weeks of gestational age with a birth weight of 2500 g (around 50th centile corrected for gestational age).

During pregnancy, her mother had regular check-ups and she did not take any medication, except for prenatal vitamins (vitamin D, vitamin B6, folic acid) and supplements (iron and calcium). In particular, the folic acid dose taken was 0.4 mg per day for one month before conception and for the first trimester of gestation.

At birth, the baby was cyanotic. Echocardiography showed complex cyanotic heart disease and she was put under inotropic support and prostaglandin infusion. On the fifth day, the baby developed jaundice and acholic stools, with a high level of conjugated bilirubin and gamma-glutamyltransferase (GGT) on blood investigations (Figure 1). The abdominal ultrasound showed mild dilation of the cholecystic duct without any changes in the intra- or extrahepatic bile ducts. The viral screening was negative for Cytomegalovirus (CMV), Rubella, Epstein–Barr virus (EBV) and Herpes simplex virus 1 (HSV1).

At 12 days of age, the proband was transferred to the Department of Pediatric Congenital Heart Disease of IRCCS Policlinico San Donato, Milan, Italy. On admission, the baby exhibited a compromised general condition with both cyanosis and jaundice. She had an oxygen saturation of 81% (environmental air), a weight of 2400 g, and a height of 49 cm. Remarkably, no hepatomegaly was described during this first clinical evaluation.

Laboratory workup revealed a high level of conjugated bilirubin (9.65 mg/dL) with a moderate increase of the transaminase level (ALT 45 U/L, AST 75 U/L), and mild increases in the C reactive protein levels (0.8 mg/L) and white cells count (23,000/mmc). The cultural exams for common pathogens were negative. The thyroid function showed TSH suppression with normal levels of fT3 and fT4. Thereafter, a genetic evaluation was recommended, noting a peculiar craniofacial appearance, characterized by a prominent forehead, deep-set eyes, bulbous nasal tip, and subtle upper lip. Taking into consideration all these clues, the geneticist raised the suspicion of Alagille syndrome.

The case was discussed with our surgical team: a congenital neural tube defect was clinically found, due to the presence of sacral dimple with an indented patch of hyperpigmented skin. Then a neuro-surgical evaluation was performed, consistent with spina bifida occulta but without indication for spinal MRI, considering both the patient’s age and her precarious clinical condition. It was felt that, in view of the marked cholestasis and severely hypoplastic pulmonary branches in the setting of Alagille Syndrome, the risk to perform palliative surgery was too high. To verify the suspicion of Alagille Syndrome, molecular analysis was performed on the *JAG1* and *NOTCH2* genes. Next-Generation Sequencing, with an Illumina-TruSight enrichment on an Illumina-NextSeq500 platform, was performed on genomic DNA extracted from a peripheral blood sample of the proband and revealed the presence of the novel *JAG1* heterozygous single-base deletion at position 802 of the cDNA (NM_000214.2:c.802delC, LOVD: https://databases.lovd.nl/shared/variants/0000579828#00000084). The presence of the variant was confirmed by Sanger Sequencing (Figure 2). No variants in the *NOTCH2* gene were detected.

To exclude the possible presence of genomic rearrangement in the patient, an Array CGH on genomic DNA extracted from peripheral blood was also performed, resulting in the discovery of a 229 kb microduplication in chromosome 17 with the following genomic position: arr [GRCh37] 17q22(55644562_55873916). This microduplication encompasses the *MSI2* gene (MIM *607897) expressing an oncogenic RNA-binding protein, required for blast crisis chronic myeloid leukemia [10]. Both parents, as well as the proband’s older brother, tested negative for both the *JAG1* variant and the microduplication within chromosome 17, suggesting that both this mutation and the genetic rearrangement are de novo (Figure 3).

Since the #182940 entry for neural tube defects in OMIM [11] describes five genes (*FUZ*, *MTHFR*, *VANGL1*, *VANGL2*, and *CCL2*) associated with neural tube defects, including spina bifida occulta, we sequenced these genes with the NGS technique. No mutation nor low-frequency variants were found, with both a medium coverage of 100X for each gene and no zone with coverage <20X Thus, the approach used for genetic testing in this proband was NGS with a panel including the genes *JAG1*, *NOTCH2*, *FUZ*, *MTHFR*, *VANGL1*, *VANGL2*, and *CCL2*. The medium coverage was 100X for each tested gene without zone with coverage less than 20X. No homozygous rare polymorphism was identified, making us confident about the absence of a deletion in all the studied genes.

In the meanwhile, the proband developed severe jaundice, including yellowing of both sclerae and lips. This prompted a further liver evaluation, revealing an increase in alanine transaminase, aspartate transaminase, glutamate-oxaloacetate transaminase, and an even larger increase in r-glutamyl transpeptidase. Conjugated “direct” bilirubin was discovered in the blood. Due to liver failure, the proband received three concentrated red blood cells transfusions and she continued on lactulose pharmaceutical therapy. The discovery of the *JAG1* variant allowed the diagnosis of Alagille syndrome, which in turn prompted further testing of thyroid, kidney, and eyes, which were all normal.

Echocardiography indicated pulmonary atresia with ventricular septal defect and severe hypoplasia of the pulmonary branches (1.5 mm), large patent ductus arteriosus and several major aortopulmonary collateral arteries (Figure 4A). In view of the hypoplasia of the pulmonary branches in a low-weight newborn, the decision was taken to perform cardiac catheterization and to implant a stent in the arterial duct. The procedure was carried out without any complications. Hypoplastic pulmonary artery branches were visualized during the catheterization that was performed before implanting the stent (Figure 4B). Eleven days after the procedure, the child was transferred back to the Romanian Hospital to continue the therapeutic management. Over the course of the next few months, the proband showed good weight gain, but with persistent cholestasis and worsening cyanosis. At three months of age, the child weighed 5 kg, with oxygen saturation between 70 and 75% and conjugated bilirubin of 10–12 mg/dL.

## 3. Discussion

It has been known since 2000 that the *JAG1* gene is expressed in normal human neural tube [12]. In spite of this, clinical descriptions of neural tube defects in molecularly confirmed Alagille patients are limited. No functional data about the role of the *JAG1* heterozygous mutation in neural tube defects are available to date. However, it has been demonstrated that apical/basal cell polarity is essential for neural tube closure [13]. In particular, it is very well known that *JAG1* is part of the NOTCH cell pathway [14] and that NOTCH signaling has an essential impact on apical/basal cell polarity [13]. The most updated data indicate that Notch signaling plays a role in both differentiation and the organization of development in the central nervous system.

To the best of our knowledge, this is the first description of a neural tube defect as an onset sign of Alagille in a pediatric patient. The vast majority of children affected by this condition display an association between cholestasis (96%) and congenital heart malformation, while the most common kind of heart involvement is pulmonic stenosis (67% of patients [15]). Spina bifida occulta has already been described in patients clinically affected by Alagille syndrome [16]. In these cases, already published, no mention about mutation(s) nor genes involved has never been provided. Instead, here we describe for the first time ever a possible cause for this phenotype, including a case of spina bifida occulta in the apparent absence of other major environmental and genetic causes. It is remarkable that the same dose of folic acid had been taken by the mother for the first trimester of gestation in the previous pregnancy, which resulted in a healthy male newborn without any clinical signs of neural tube defect.

In the present case, the clinical picture was characterized at its onset by congenital spina bifida occulta and cyanotic heart disease: this is not a common association to lead the pediatrician to the suspicion of Alagille syndrome. This condition is more commonly suspected due to the association between cholestasis and congenital heart malformations.

The face of a patient is an important element, as well, to raise the clinical suspicion: it is a triangular-shaped face with a prominent forehead, hypertelorism, deep-set eyes, a pointed chin, and a bulbous nasal tip. Most of these features are actually present in our patient as well, and they are one of the elements that justified genetic testing as a diagnostic approach.

Alagille syndrome is characterized by a marked variability, even in the same family: some patients could display a serious clinical picture from the first hours of life (often due to cardiac congenital malformation) or early infancy. On the other hand, some patients might suffer from apparently isolated cholestasis of unknown cause or even have only a particular ocular malformation [17].

*JAG1* is the major gene causing Alagille syndrome, [18] with more than 600 pathogenic variants identified to date [7]. These variants are known to be truncating in the majority of cases (49% frameshift, 16% nonsense, 9% splicing [15], 11% large gene rearrangements), and non-truncating in about 15% (missense) [15]. In 1–2% of affected patients, Alagille syndrome is caused by heterozygous pathogenic variants in *NOTCH2* on chromosome 1p13.2. No pathogenic variants are identified in about 4% of affected patients, suggesting that some other genes might play a causative role for this subset of cases [19].

To date, there is not a clear correlation between genotype and phenotype, and no clinical pictures seem exclusively associated with *JAG1* or *NOTCH2*. Both Jagged1 and NOTCH proteins belong to the Notch pathway, a highly conserved signaling system expressed in many multicellular organisms and specifying cell fate decisions [20]. In humans, Notch signaling is based on four different proteins (NOTCH1-4) that function all as receptors interacting with a family of transmembrane ligands expressed on the surface of adjacent cells. Among these, Jagged1 is the ligand of NOTCH1. Recent studies resolved the structure of the Notch-Jagged extracellular interacting complex, revealing a binding interface. Luca and coauthors demonstrated how this interaction is specific for binding, ligand discrimination, and Notch signaling [21]. This relationship regulates many development processes (in particular pulmonary veins, bile ducts, pancreas, and kidneys).

Skeletal development requires normal Notch signaling. In fact, skeletal defects are features shared between Alagille and Hajdu-Cheney syndromes [22]. According to current knowledge, a whole gene heterozygous deletion (regardless of whether in *JAG1* or *NOTCH2*) does not provoke a somite-related phenotype, suggesting a role for Notch signaling in the early stages of osteogenesis [23]. This might be the reason why some Alagille patients are affected by the butterfly vertebrae, a persistent notochordal tissue between the anterior vertebral arches [24].

Our patient is the first case of Alagille syndrome with the novel heterozygous single nucleotide deletion at position 802 (c.802del) of the *JAG1* gene (reference sequence NM_000214.3), resulting in frameshift and insertion of a premature stop of translation after 144 codons, in the putative protein (p.His268Thrfs*144). This specific truncating variant has not been reported to date in patients clinically affected by Alagille syndrome and involves a highly conserved region of the gene (GERP score 5.92). It is possible that the heterozygous state for this mutation explains the entire phenotype of our proband, because it is not present in the general population nor in the healthy parents (*de novo* hypothesis), the resulting putative JAG1 protein is truncated, and the patient does not carry any mutation in the currently known neural tube defect genes, according to OMIM [11].

The c.802delC variant causes the substitution of His268, a residue anchoring the interface between NOTCH1-EGF10 and Jag1-EGF1 domains, as recently described [21]. We do not know if the truncated protein is present or degraded, but evidence exists that the mutated region might be critical for the Notch-Jagged1 complex. Functional studies are required to confirm this hypothesis.

In addition, it is very well known that heterozygous frameshift mutations in either *JAG1* or *NOTCH2* genes cause Alagille phenotypes similar to the one displayed by our patient [25,26]. Variants in the *NOTCH2* gene have been associated with neural tube defects [27], but the proband in this report was not found to have a *NOTCH2* variant, because the massively parallel sequencing of this gene demonstrated a wild type sequence. This is an important clue suggesting that, in addition to heterozygous deletions in *JAG1* and *NOTCH2* described in other studies, the heterozygous state for the *JAG1* mutation described in this case alone might be independently responsible for neural tube defects. We cannot exclude, however, that other genetic variants might have caused the neural tube defect, even in the presence of evidence that the Jagged 1 pathway is actually involved in the pathogenesis of such a condition.

It is remarkable that in the Online Mendelian Inheritance in Men website [11], there are some rare cases described of neural tube defects (even more serious than in our patient) caused by a heterozygous mutation in a gene that belongs to the Notch pathway. In particular, one study described a few patients [28] with neural tube defects harboring heterozygous mutations in the *NOTCH3* gene. The protein codified by this gene shows similarity with *NOTCH2* [29], so it is possible that neural tube defects might be caused also by mutations in other genes coding for proteins that belong to the Notch signaling pathway. This mechanism had been already described in many other genetic conditions such as Rasopathies [30], coesinopathy [31], and mTOR related conditions [32]. Consequently, other genes associated with neural tube defects [11], including spina bifida, were sequenced, with negative results.

In conclusion, we have to mention that to reach a better diagnostic classification, a CGH Array analysis from peripheral blood extracted genomic DNA was performed. The main reason is that, in some patients affected by neural tube defects, a submicroscopic genomic rearrangement has been demonstrated [33]. Our patient harbors a microduplication in the long arm of chromosome 17 with unknown inheritance, identified with the CGH Array technique. This microduplication encompasses the *MSI2* gene (MIM *607897) known to be involved in TGF beta signaling, but not to date in the Notch pathway [34]. To date, no *MSI2* mutation, deletion, nor duplication has been described in patients with neural tube defects. Taken together, these two statements strengthen our hypothesis that the *JAG1* frameshift variant demonstrated in our patient might be considered at least the major cause of her phenotype. We recommend that Alagille syndrome should be considered as a possible diagnosis in the presence of congenital heart malformations and neural tube defects, even in the absence of liver involvement. Future studies are needed to understand this relationship further. In particular, it needs to be assessed whether neural tube defects might be present also in other conditions (for example Hajdu Cheney syndrome) caused by heterozygous mutations in genes coding for Notch pathway proteins.

## 4. Concluding Remarks

This work demonstrates a novel pathogenic heterozygous *JAG1* mutation is associated with an atypical Alagille phenotype. The clinical picture of this female patient is characterized by some common Alagille signs associated with a congenital spina bifida occulta. This association has never been described before to the best of our knowledge: the keywords “Alagille” and “spina bifida occulta” provide no result in Pubmed (https://www.ncbi.nlm.nih.gov/pubmed/?term=Alagille+AND+spina+bifida+occulta, as of October 21, 2019). Considering the major role of Notch signaling in neural tube defect pathogenesis, our case suggests that Alagille patients might have an increased risk for neural tube defects compared to the general population. However, further studies are required to confirm this finding.

## Figures and Tables

**Figure 1 ijms-20-06247-f001:**
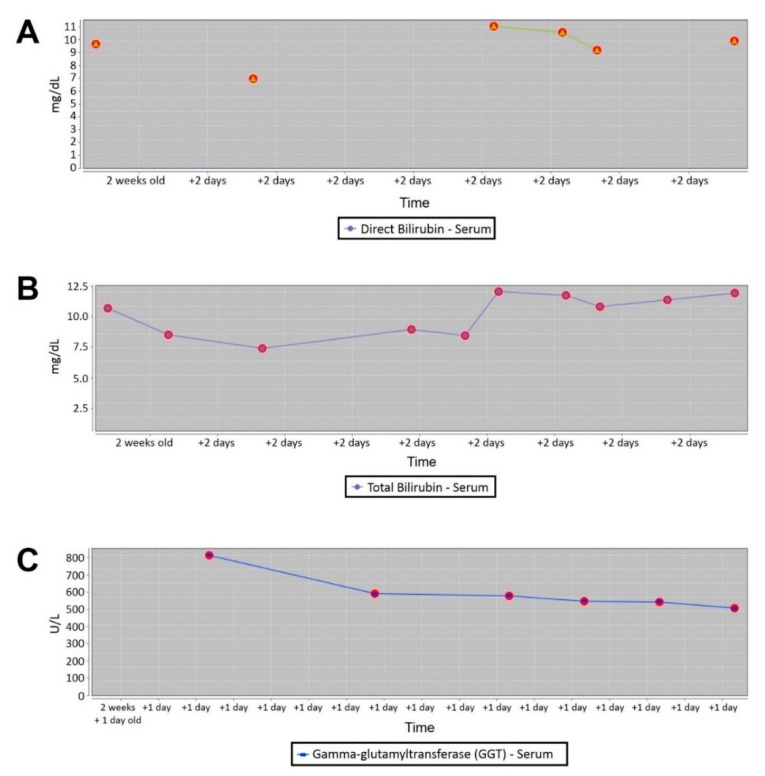
(**A**) Serum direct bilirubin concentration. (**B**) Serum total bilirubin concentration. (**C**) Serum gamma-glutamyltransferase concentration.

**Figure 2 ijms-20-06247-f002:**
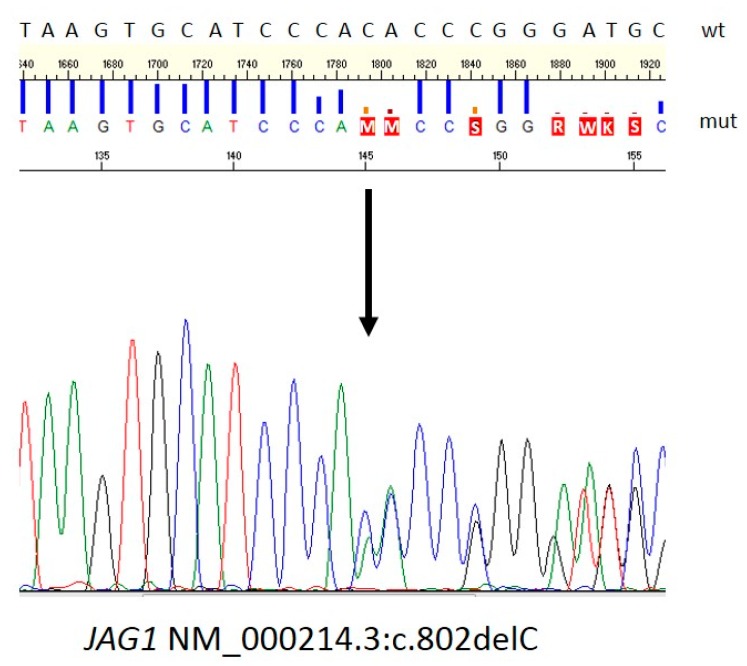
Sanger direct sequencing of c.802delC. The variation creates a frameshift at codon His268. The new reading frame ends in a stop codon at position 144; wt: wild type; mut: mutant. The letters with red box means the called basis when there is a frameshift in the called position. So the letters with red box must be read as follows: M = C or A, S= C or G, R= G or A, W= T or A, K = G or T.

**Figure 3 ijms-20-06247-f003:**
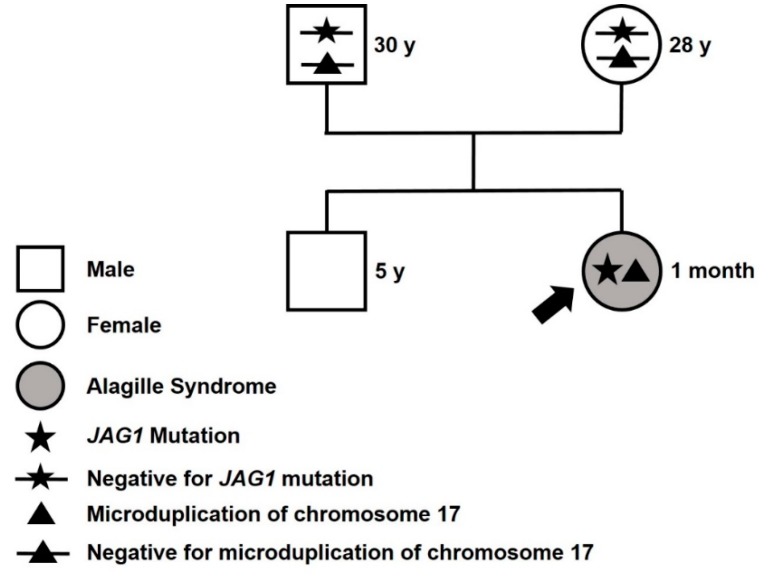
Family pedigree. Proband identified with an arrow. y = years old at diagnosis.

**Figure 4 ijms-20-06247-f004:**
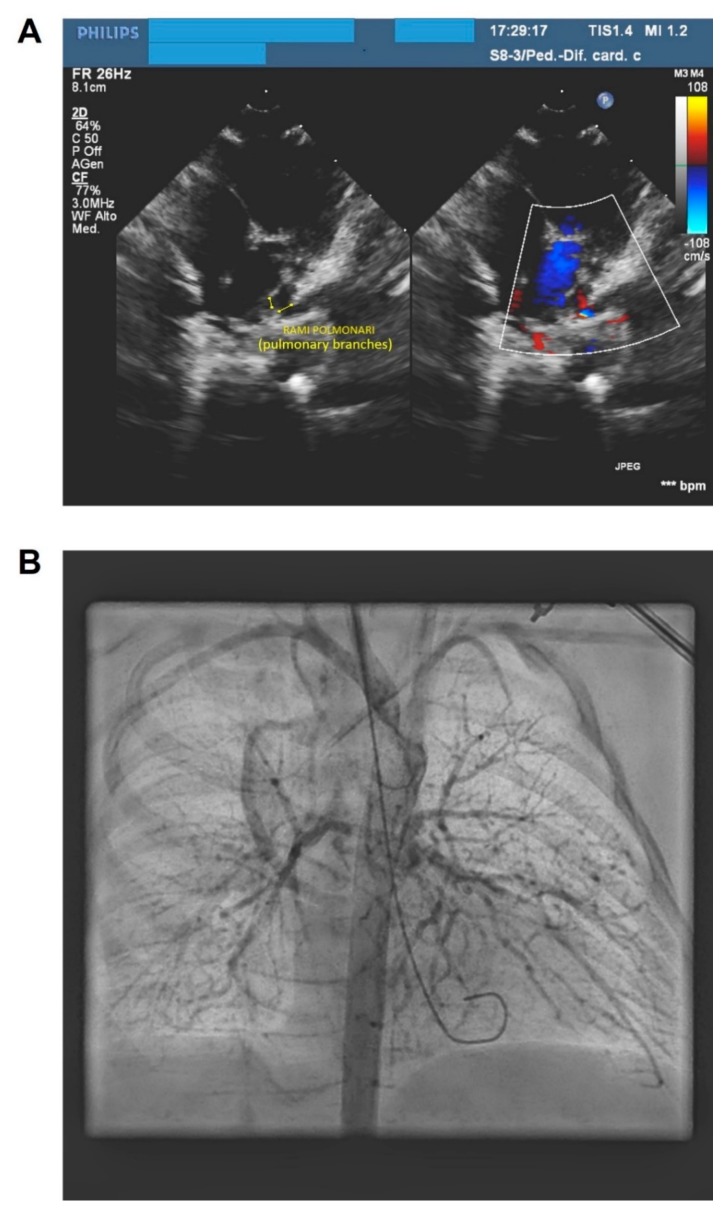
(**A**) Echocardiography indicated severe hypoplasia of the pulmonary branches. (**B**) Hypoplastic pulmonary artery branches were visualized during the catheterization that was performed before implanting the stent.

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
