# Peer review of "Novel JAG1 Deletion Variant in Patient with Atypical Alagille Syndrome"

_ijms, 2019, doi:10.3390/ijms20246247_

Round 1
Reviewer 1 Report
The authors present the case of a child exhibiting a phenotype concordant with Alagille syndrome. Among the symptoms, the presence of a congenital neural tube defect is attributed to a novel mutation in the JAG1 gene that produces a truncated protein.
The clinical presentation of the case is well done and all the necessary information is provided. However, the evidence provided to support the idea that the NTD is solely due to the mutation in the JAG1 gene is weak.
First of all, the description of a novel mutation in the JAG1 gene leading to a truncated protein is interesting but this is also true for typical mutations leading to AGS.
Second, in order to conclude that the NTD is caused solely by the JAG1 mutation, more experiments need to be done, for example, the rest of the genome should be investigated looking for variants with possible implications in NTD.
Due to these reasons, I would recommend either the paper to be rewritten to be a report case of Alagulle syndrome but without any connection between the mutation and the NTD or to do a more thorough investigation of the patient's genome to be able to establish a more convincing connection between this variant and the neural tube defect.
Author Response
We believe that it is true that no functional data about the JAG1 heterozygous mutation are available to date. However, it has been demonstrated that apical/basal cell polarity is essential for neural tube closure (Main et al, 2013. PMID 23675446). In particular, it is very well known that JAG1 is part of the NOTCH cell pathway (Grochowski et al, 2016. PMID 26548814) and NOTCH signaling has an essential impact on apical/basal cell polarity (Main et al, 2013. PMID 23675446). The most updated data indicate that Notch signaling plays a role in both differentiation and organization of development in the central nervous system. “Spina bifida occulta” has already been described in patients clinically affected by Alagille syndrome (Penton et al, 2012. PMID 22306179). In these cases, already published, no mention about mutation(s) nor genes involved had been provided. Instead, we are describing for the first time ever a possible cause (even if not confirmed) for this phenotype, including a case of spina bifida occulta in the apparent absence of other major environmental and genetic causes.
In fact, the # 182940 entry for neural tube defects in OMIM describes five genes (FUZ, MTHFR, VANGL1, VANGL2 and CCL2) associated with neural tube defects, including spina bifida. We sequenced these genes with the NGS technique in response to the reviewer comments. No mutation nor low frequency variants were found, with both a medium coverage of 100X for each gene and no zone with coverage <20X. Thus, the approach used for genetic testing in this proband was thus NGS with a panel including the genes JAG1, NOTCH2, with the subsequent addition of FUZ, MTHFR, VANGL1, VANGL2 and CCL2 after the reviewer comments. The medium coverage was 100X for each tested gene without zone with coverage less than 20X. No homozygous rare polymorphism have been identified, making us confident about the absence of deletion in the studied genes.
We are aware that many other genes, still unknown, might be associated with neural tube defect, but this is true also for Alagille syndrome and in general for each genetic condition. Further studies, especially from a functional point of view, are warranted to assess our hypothesis about the mechanism of neural tube defect in Alagille syndrome.
Reviewer 2 Report
Pappone and colleagues report a rare case of a JAG1 mutation causing Allagille syndrome with neural tube defect.
The case is unique and written well but the weakness of the report is a missing functional analysis making the conclusion that the observed neural tube defect is linked to this mutation highly speculative and not conclusive.
Major points:
Page 2 line 65 you mention, that the mother prenatally consumed folic acid. As this is indicated to reduce the risk for both neural tube defects and heart defects even before pregnancy please provide the concentration and duration of the supplement uptake. You define the two patient mutations (Jag1 and dup17) as de novo, but there is no statement if paternity is tested. This is the first and only Alagille syndrome case with JAG1 mutation and additional neural tube defect. Besides that this could just be coincidence there are no experimental or even literature data supporting a causative connection! Page 7 lines 188-189 ‘We do not know if the truncated protein is present or degraded, but evidence exists that the mutated region is critical for the Notch-Jagged1 complex.’ And page 8 lines 194-197 ‘This is an important clue suggesting that, in addition to heterozygous deletions in JAG1 and NOTCH2 described in other studies, the heterozygous state for the JAG1 mutation described in this case alone might be independently responsible for neural tube defects.’ So again this is highly speculative! No experiments and no reference is given!
Minor points:
Add OMIM reference numbers for AGS. Figure 1A single measuring points are missing and as the graph is somehow similar to 1B I suggest skipping Figure 1A. Page 3 line 95 you mention NGS which strategy was applied-multi-gene panel, WES? Page 3 lines 102-103 the provided arrayCGH formula is not according to ISCN2016 which is also not referenced. Figure 3 the letters in the red labeled mutant allele are not readable.
Author Response
We believe that it is true that no functional data about the JAG1 heterozygous mutation are available to date. However, it has been demonstrated that apical/basal cell polarity is essential for neural tube closure (Main et al, 2013. PMID 23675446). In particular, it is very well known that JAG1 is part of the NOTCH cell pathway (Grochowski et al, 2016. PMID 26548814) and NOTCH signaling has an essential impact on apical/basal cell polarity (Main et al, 2013. PMID 23675446). The most updated data indicate that Notch signaling plays a role in both differentiation and organization of development in the central nervous system. “Spina bifida occulta” has already been described in patients clinically affected by Alagille syndrome (Penton et al, 2012. PMID 22306179). In these cases, already published, no mention about mutation(s) nor genes involved had been provided. Instead, we are describing for the first time ever a possible cause (even if not confirmed) for this phenotype, including a case of spina bifida occulta in the apparent absence of other major environmental and genetic causes.
In fact, the # 182940 entry for neural tube defects in OMIM describes five genes (FUZ, MTHFR, VANGL1, VANGL2 and CCL2) associated with neural tube defects, including spina bifida. We sequenced these genes with the NGS technique in response to the reviewer comments. No mutation nor low frequency variants were found, with both a medium coverage of 100X for each gene and no zone with coverage <20X. Thus, the approach used for genetic testing in this proband was thus NGS with a panel including the genes JAG1, NOTCH2, with the subsequent addition of FUZ, MTHFR, VANGL1, VANGL2 and CCL2 after the reviewer comments. The medium coverage was 100X for each tested gene without zone with coverage less than 20X. No homozygous rare polymorphism have been identified, making us confident about the absence of deletion in the studied genes.
We are aware that many other genes, still unknown, might be associated with neural tube defect, but this is true also for Alagille syndrome and in general for each genetic condition. Further studies, especially from a functional point of view, are warranted to assess our hypothesis about the mechanism of neural tube defect in Alagille syndrome.
We have incorporated these concepts and references throughout the manuscript.
Regarding the folic acid, in the manuscript, we now write, “…the folic acid dose taken was 0.4 mg per day for one month before conception and for the first trimester of gestation.” We also write, “It is remarkable that the same dose of folic acid had been taken by the mother for the first trimester of gestation in the previous pregnancy. However, that previous pregnancy resulted in a healthy male newborn without any clinical signs of neural tube defect.”
Regarding the CGH, we write, “To exclude the possible presence of genomic rearrangement in the patient, an Array CGH on genomic DNA extracted from peripheral blood was also performed, resulting in the discovery of a 229 kb microduplication in chromosome 17 with the following genomic position: arr [GRCh37] 17q22(55644562_55873916). This microduplication encompasses the MSI2 gene (MIM *607897) expressing an oncogenic RNA-binding protein, required for blast crisis chronic myeloid leukemia [11].”
We have modified the letters of the mutant allele in the figure to be white, so that they can stand out better against the dark red background.
Round 2
Reviewer 1 Report
The additional experiments and changes to the text in the manuscript make it more balanced and ready for publication.
Reviewer 2 Report
The author improved the manuscript according to reviewer comments. Although the experimental prove is not provided the discussion now includes this points and make it more clear.
The manuscript is now ready to be published.